# Description of maternal and neonatal adverse events in pregnant people immunised with COVID-19 vaccines during pregnancy in the CLAP NETWORK of sentinel sites: nested case–control analysis of the immunization-associated risk – a study protocol

Diego Macías Saint-Gerons ![ORCID],[1] José Luis Castro ![ORCID],[2] Mercedes Colomar ![ORCID],[3,4] Robin Rojas-Cortés ![ORCID],[5] Claudio Sosa ![ORCID],[6] Alba Maria Ropero,[7] Suzanne Jacob Serruya ![ORCID],[8] Desiré Pastor,[7] Monica Chiu,[7] Martha Velandia-Gonzalez,[7] Edgardo Abalos ![ORCID],[9] Pablo Durán ![ORCID],[8] Rodolfo Gomez Ponce de León ![ORCID],[8] Giselle Tomasso ![ORCID],[3] Luis Mainero,[8] Marcelo Rubino,[8] Bremen De Mucio ![ORCID] [8]

**Correspondence to**
Dr Mercedes Colomar;
mcolomar@unicem-web.org

## ABSTRACT

**Introduction** COVID-19 is associated with higher morbimortality in pregnant people compared with non-pregnant people. At present, the benefits of maternal immunisation are considered to outweigh the risks, and therefore, vaccination is recommended during pregnancy. However, additional information is needed on the safety of the vaccines in this population.

**Methods and analysis** This a retrospective cohort nested case–control study in pregnant people who attended maternity hospitals from eight Latin American and Caribbean countries. A perinatal electronic clinical history database with neonatal and obstetric information will be used. The proportion of pregnant people immunised with COVID-19 vaccines of the following maternal and neonatal events will be described: preterm infant, small for gestational age, low birth weight, stillbirth, neonatal death, congenital malformations, maternal near miss and maternal death. Moreover, the risk of prematurity, small for gestational age and low birth weight associated with exposure to COVID-19 vaccines will be estimated. Each case will be matched with two groups of three randomly selected controls. Controls will be matched by hospital and mother's age (±3 years) with an additional matching by delivery date and conception time in the first and second control groups, respectively. The estimated required sample size for the main analysis (exposure to any vaccine) concerning 'non-use' is at least 1009 cases (3027 controls) to detect an increased probability of vaccine-associated event risk of 30% and at least 650 cases (1950 controls) to detect 30% protection. Sensitivity and secondary analyses considering country, type of vaccine,

## STRENGTHS AND LIMITATIONS OF THIS STUDY

⇒ Pregnant people were excluded from the pivotal clinical trials of COVID-19 vaccines and most of the evidence from vaccine safety comes from high-income country studies.

⇒ This study represents an opportunity to generate complementary evidence of maternal, fetal and neonatal adverse events after administration of COVID-19 vaccines currently used in Latin American and Caribbean countries for maternal immunisation.

⇒ The use of the SIP-PLUS database allows achieving a large sample size necessary to explore potential vaccine-associated discrete effects (beneficial or harmful) in a timely manner.

⇒ As in any study using routinely collected data, there are further potential limitations due to incomplete data records, inaccuracies (dates), unmeasured confounding factors and missing data.

⇒ The number of 'non-exposed' pregnant people could decrease over time and may affect statistical power and interpretation.

exposure windows and completeness of immunisation will be reported.

**Ethics** The study protocol was reviewed by the Ethical Review Committee on Research of the Pan American Health Organization. Patient informed consent was waived due to the retrospective design and the utilisation of anonymised data (Ref. No:

PAHOERC.0546.01). Results will be disseminated in open access journals.

## INTRODUCTION

Pregnant people do not appear to have a higher susceptibility to SARS-CoV-2 coronavirus infection than non-pregnant people.[1] However, COVID-19 is associated with higher morbidity and mortality among pregnant people compared with non-pregnant people.[2 3] Mortality in pregnant people with COVID-19 could be 70% higher, and the risk of admission to intensive care units (ICUs) or the need for mechanical ventilation would be three times higher compared with non-pregnant unvaccinated people of similar age.[3 4] Pregnant people with COVID-19 show an increased risk of occurrence of some adverse maternal–fetal outcomes such as increased risk of preterm delivery, pre-eclampsia or miscarriage compared with pregnant people without COVID-19.[2] It has been observed that, in its most complicated or critical forms, COVID-19 infection is associated with prematurity, gestational diabetes, pre-eclampsia and low birth weight.[3] Neonates born to mothers with COVID-19 appear to have a higher probability of admission to a neonatal care unit, which could be attributed to a greater extent to the negative effect of COVID-19 on maternal health during gestation than to vertical transmission.[4–6] In addition, other maternal, fetal and neonatal complications associated with COVID-19 have also been reported. A systematic review described adverse events such as fetal distress, premature rupture of membranes and other neonatal adverse events such as respiratory distress, gastrointestinal events and fever. Maternal death reports have been more rare.[7]

After a global research effort to develop effective treatments and vaccines against COVID-19, vaccines became available on January 2021. It should be noted that a characteristic of the new COVID-19 vaccines is that they were developed from a range of different platforms: viral vector (adenoviral), mRNA platform, inactivated, attenuated viruses, being pregnancy an exclusion criterion in the main pivotal clinical trials undertaken for the authorisation of these COVID-19 vaccines.[8] In addition, vaccines that are currently recommended for maternal immunisation—and with which there is most experience—such as the influenza vaccine or the Tdap vaccine, do not use mRNA-based platforms as is the case with the Pfizer-BioNTech and Moderna vaccines.[9] However, despite being based on a newer platform, mRNA vaccines do not entail the risk of acquiring vaccination-induced infection, as might be the case with the use of attenuated/inactivated vaccines. On the other hand, they could be useful in pregnant people, as a robust transfer of the humoral response to the fetus has been observed with mRNA vaccines.[10 11] Furthermore, no remarkable adverse events have been detected in experimental genotoxicity studies conducted on rats.[12]

Although various clinical trials have been planned for the study of COVID-19 vaccines in pregnant people, most of the initiatives are based on the creation of registries of pregnant people exposed to the vaccines. In many cases, it is the adaptation of previous registries that had been created to study the effects of COVID-19 infection in pregnant people. In this respect, Moderna created a registry to monitor pregnancy outcomes in individuals who received their COVID-19 vaccine. PREGISTRY—for example—is a startup focused on improving maternal–fetal health, in collaboration with the Harvard School of Public Health (https://www.pregistry.com); aim to enrol 8000 pregnant people in their registry to evaluate their obstetric, neonatal and infant outcomes following COVID-19 immunisation. In the UK, the UKOSS registry (https://www.npeu.ox.ac.uk/ukoss) is collecting data to allow the parallel nesting of cohort and case–control studies. The CDC's 'v-safe' registry has reported preliminary results with observed frequencies of spontaneous abortion and fetal death in vaccinated pregnant people similar to the expected frequencies in the general population of pregnant people of the same age and without known comorbidities.[13]

Apart from these active registries that include pregnant people, there is little evidence from published studies with complete data on the effectiveness and safety of vaccination for COVID-19 in this population and information from surveillance systems, monitoring acute adverse events following immunisation during pregnancy, did not find increased risk of clinically serious acute adverse in different populations groups.[14 15] One retrospective cohort study observed that, in pregnant people, vaccination with Pfizer-BioNTech was associated with a lower risk of SARS-CoV-2 infection compared with those who did not receive vaccination (HR 0.22; 95% CI 0.11 to 0.43).[16] In another prospective cohort study, short-term adverse events associated with mRNA vaccines were similar among pregnant people to those occurring in non-pregnant people of the same childbearing age.[17] In another large comparative study involving 157 521 singleton pregnancies from the Swedish pregnancy and Norwegian childbirth electronic registries, no statistically significant differences were found in the risk for fetal death, small for gestational age rates, preterm and low Apgar score (Appearence, Pulse, Grimace, Activity and Respiration) value in mRNA and Pfizer-BioNTech vaccinated and unvaccinated subjects.[18] Also, the risk of miscarriage does not appear to be higher than expected among people immunised before or during pregnancy.[19 20] Moreover, in a retrospective cohort including more than 40 000 pregnant people, COVID-19 vaccines (mostly mRNA) were not associated with preterm birth or small-for-gestational-age at birth.[21] On the other hand, a case–control study using Norwegian registry data observed no statistically significant risk difference for first-trimester miscarriage in pregnant people exposed to COVID-19 vaccines (Pfizer-BioNTech, Moderna, Astra) and those not immunised.[22] A study investigating on the associations of COVID-19 vaccination throughout pregnancy with delivery and neonatal outcomes suggests that there is no associated risk between prenatal COVID-19

vaccination and adverse delivery and neonatal outcomes in a cohort sample from NYC.[23] A systematic review and meta-analysis included data from 23 studies and 117552 pregnant people immunised mostly with mRNA vaccines. In this meta-analysis, which accumulates the most evidence on the subject to date, no differences were found in vaccinated compared with unvaccinated pregnant people with regards the risk of occurrence of miscarriage, earlier gestation at birth, placental abruption, pulmonary embolism, postpartum haemorrhage, maternal death, mothers' admission to the ICU, low birth weight or newborns' admission to the ICU. The meta-analysis identified that immunisation conferred a 15% protective effect against fetal death, although the heterogeneity found was very high (OR 0.85; 95%CI 0.73 to 0.99; $I^2$=93.9%).[24] Some absolute estimates of the impact of vaccination with mRNA vaccines have been proposed to assess the number of maternal and neonatal adverse outcomes that could be avoided such as 'number needed to vaccinate (NNV)'. The impact measures suggest a benefit in avoiding certain events such as COVID-19 infection (NNV:11), caesarean section delivery (NNV: 182) and prematurity (NNV: 200), while the NNV would be higher for 'neonatal problems' (NNV: 463), severe cases of COVID-19 (NNV: 412–2058), requirement for mechanical ventilation (NNV: 1371–6857) and small for gestational age (NNV: >2000). In all cases far from the 'number needed to harm' (NNH) estimated for some adverse events: myocarditis (NNH: 37 000) or thrombosis syndrome with thrombocytopenia (NNH: 50 000).[25] estimates carried out based on observational data in Israel to determine the 'number needed to boost' and assess the impact of the third booster dose of the Pfizer-BioNTech vaccine in preventing hospital admissions for COVID-19 and complicated COVID-19 would also suggest a greater potential benefit for pregnant people of advanced age or with comorbidities receiving booster doses.[26]

However, majority of evidence suggesting the possible beneficial effect of vaccination comes from descriptive observational studies. A large population-based descriptive study included more than 180 000 pregnant people in Scotland. The results suggested a possible benefit of vaccination. Thus, 77.4% of all COVID-19 infections, 90.9% of hospital admissions, 98% of ICU hospitalisations and 100% of neonatal deaths occurred in pregnant people diagnosed with COVID who had not been vaccinated.[27] In addition, descriptive data from six European countries from the International Network of Obstetric Survey Systems have also reported that the highest proportion of ICU admissions of pregnant people corresponded to those who had not been vaccinated (80%–100% of the total).[28] On the other hand, in a study in the USA from electronic medical record data in women who had miscarriages, no differences were found in the probability of having been vaccinated (mRNA vaccines) 28 days before the occurrence of the miscarriage.[29]

At present, the benefits of maternal immunisation are considered to outweigh the risks and for that reason

vaccination is recommended at any time during pregnancy.[1 30 31] The recommendation is supported, on the one hand, by the robust evidence indicating a higher morbidity and mortality associated with COVID-19 among pregnant people compared with non-pregnant people; and on the other hand, by the protective effect of maternal immunisation observed against complications of COVID-19. Moreover, no pharmacovigilance signals associated with maternal immunisation have been detected[13 14 32 33] Finally, some studies suggest a decreased risk of occurrence for certain adverse fetal and neonatal outcomes associated with maternal immunisation, although these are considerations based mostly on observational evidence, and in many cases with the caveat that further studies are needed. However, the great difference in the amount of evidence available according to the type of vaccine is striking. Most studies in pregnant people have been carried out with vaccines developed from the newest mRNA platform (Pfizer-BioNTech, Moderna), with very little evidence available for the other vaccines. Perhaps for this reason, some countries prioritise the use of mRNA vaccines in their recommendations, leaving the administration of other (non-mRNA) vaccines with a weaker recommendation based on clinical criteria, weighing the possible risk of infection of the mother considering certain circumstances (eg, health workers, comorbidities) with the theoretical risks that the uncertainties of immunisation may entail.[34 35] In spite of the available evidence, most countries that have reported vaccination coverage in pregnant people had low vaccinations rates.[36] For example, in Scotland, only 32.3% of pregnant people had received the full scheme (vs 77.4% of non-pregnant people of childbearing age).[27] Other countries have also reported low coverage rates among pregnant people such as Italy (20%) or England (22%) or 30%–50% in the Netherlands.[28]

The Latin American Center of Perinatology (CLAP) coordinates a NETWORK of sentinel centres in Latin American and Caribbean countries for the surveillance of maternal health related issues (see table 1). All centres use the Perinatal Information System (SIP, for its acronym in Spanish) on its plus version (SIP-PLUS) as a common data collection system. SIP-PLUS database contains the perinatal electronic health record data (online supplemental material 1). The perinatal medical record prospectively and systematically records individual data on the clinical follow-up of pregnant people seen from the date of the first antenatal visit to the date of discharge following delivery. It is a healthcare database of perinatal electronic medical records that centralises the demographic, diagnostic, obstetric and neonatal clinical data of mothers and their offspring. In addition, it incorporates the systematic collection of information on the use of vaccines (tetanus/diphtheria, Tdap, influenza, rubella, hepatitis B, hepatitis A), as well as information on dosage, date of vaccination and time of vaccination (before or during pregnancy, or after delivery).[37] In response to the outbreak of the COVID-19 pandemic, CLAP developed a

**Table 1** Characteristics of the hospitals that use the SIP-PLUS database

| Country | Income level | Name of hospital | Level | Deliveries attended per year |
|---|---|---|---|---|
| Argentina | UMIC | Martin Maternity Hospital, Rosario | Tertiary | 3600 |
| | | Roque Sáenz Peña, Rosario | | 2500 |
| | | Eva Perón, Rosario | | 1500 |
| Bolivia | LMIC | Bolivian-Japanese Hospital | Tertiary | 2719 |
| | | La Paz Maternity Hospital | Tertiary | 1500 |
| Colombia | UMIC | Rafael Calvo Maternity Clinic | Secondary | 7000 |
| | | Hospital Universitario del Valle, Cali | Tertiary | 7200 |
| Ecuador | UMIC | Isidro Ayora Maternity Hospital | Tertiary | 7000 |
| Honduras | LMIC | Roberto Suazo Cordova Hospital | Secondary | 3201 |
| | | San Felipe Hospital | Secondary | 6128 |
| | | Leonardo Martínez Valenzuela | Tertiary | 18000 |
| Nicaragua | LMIC | España Hospital | Secondary | 5250 |
| | | Berta Calderón Roque Hospital | Secondary | 8993 |
| Dominican Republic | UMIC | San Lorenzo de Los Mina Hospital | Tertiary | 13000 |
| Uruguay | UMIC | Pereira Rosell Hospital Center | Tertiary | 5000 |
| | | Hospital de Clinicas Manual Quintela | Tertiary | 800 |

LMIC, low-income and middle-income country; SIP PLUS, Perinatal Information System (SIP, for its acronym in Spanish) on its plus version (SIP-PLUS) ; UMIC, upper-middle-income country.

specific SIP form to monitor COVID vaccination of pregnant people, along with other related variables, such as COVID-19 history, which have become part of the routine follow-up of pregnant people (online supplemental material 2).

Therefore, this project aims to retrospectively describe the use of COVID-19 vaccines among pregnant people and explore the risk associated with vaccination and selected adverse events, adapting the SIP-PLUS database for surveillance of the use and safety of COVID-19 vaccines from data routinely collected during obstetric and neonatal monitoring of pregnant people in a network of Latin American centres.

## GENERAL AND SPECIFIC OBJECTIVES
### General objective
To describe the utilisation of COVID-19 vaccines by pregnant people in hospitals using the SIP-PLUS database and to analyse the risk of vaccination associated with selected maternal and neonatal events.

### Specific objectives
1. To describe the proportion of vaccinated individuals among pregnant people by type of COVID-19 vaccine, number of doses administered and country.
2. Describe the frequency (cumulative incidence) of the following neonatal and maternal events in the cohort of pregnant people: preterm delivery, small for gestational age, low birth.
3. To analyse the associated risk of vaccination for the following neonatal events of interest through an exploration of case–controls nested in a cohort for the

following events: prematurity, small for gestational age and low birth weight.

## METHODS AND ANALYSIS
### Design and rationale
Retrospective observational descriptive cohort and analytical study with a nested case–control approach. All outcome and exposure data will be available before the study start date (retrospective analysis).

The dedicated module for capturing COVID-19 vaccine exposure information (including possible heterologous combinations) will be used to obtain the data required. This module complements the current data routinely collected for the vaccines recommended during pregnancy and is collected as part of the obstetric clinical monitoring (online supplemental material 2).

### Source of information and study period
This study includes secondary data. All the necessary information, including sociodemographic data, comorbidities, exposure to vaccines and events, will be retrieved from the SIP-PLUS perinatal electronic medical record database that electronically consolidates perinatal medical record data obtained during routine clinical practice over a retrospective 12-month period.

### Population, outcome variables, exposure and others
*Retrospective description of vaccination and event frequency in the pregnant people cohort*
*Inclusion criteria*
All pregnant people admitted for delivery (singleton) or miscarriage from October 2022 to July 2023.

## Exclusion criteria

Twin pregnancies will be excluded.

## Adverse events of interest

1. Premature births (preterm delivery)
   Born alive at less than 37 weeks gestational age as registered in the SIP form clinical record.
2. Small for gestational age. Live birth classified as small by country standards.
3. Low birth weight. Born alive weighing less than 2500 g.
4. Stillbirth (at any time).
5. Non-extended neonatal death (max 48 hours post partum).
6. Congenital malformations (major and minor).
7. Maternal death (at any time).
8. Maternal near miss (MNM). Pregnant people with an MNM condition according to WHO MNM approach.[38]
9. Abortion (spontaneous abortion and therapeutic abortion).

## Covariates

Sociodemographic variables:
► Mother's age (years).
► Ethnicity (white, indigenous, mixed, other).
► Marital status (married, common-law marriage, single, other).
► Educational level (none, elementary, high school, college, university).
► Country
► Hospital.
   Obstetric variables and risk factors (definitions)
► Date of last menstrual period (LMP).
► Gestational age (weeks)
► Smoking (yes/no).
► Drug use (yes/no).
► Alcoholism (yes/no).
► Induced labour (yes/no).
► Diabetes
   – Pre-existing (yes/no)
   – Gestational (yes/no).
► High blood pressure
   – Pre-existing (yes/no)
   – Gestational (yes/no).
► Eclampsia (yes/no).
► Pre-eclampsia (yes/no).
► Parity (nulliparous, multiparous).
► Intergestational period (months).
► COVID-19 infection (describe diagnosis, prior to or during pregnancy).
   Administration of other recommended vaccines during pregnancy
► Tetanus/diphtheria (yes/no).
► Tdap (yes/no)
► Hepatitis B (yes/no).
► Influenza (yes/no).
► Any of the other recommended vaccines (yes/no).

## Exposure of interest (immunisation with COVID-19 vaccines)

Type of vaccine, number of doses administered and date. The vaccines of interest will be[33]
► Astra-Zeneca (AZD1222; ChAdOx1 nCoV-19; ChAdOx1-S; Vaxzevria, Covishield).
► Janssen (NJ-78436735; Ad26.COV2-S).
► Moderna (mRNA-1273; Elasomeran).
► Pfizer-BioNTech (BNT162b2; Tozinameran; Comirnaty).
► Sinopharm (Sinopharm/BIB).
► Sinovac (CoronaVac; VeroCell).
► Sputnik-Gamaleya (Gamaleya; Gam-COVIDVac; Adeno-based (rAd26-S+rAd5 s).

The rest of vaccines not considered in the list will be recorded in free text as 'other vaccines'. When the administration of two different vaccines is included, the vaccination will be classified as 'heterologous schedule'. On the other hand, it will be considered a complete schedule if the pregnant person received two doses (or one dose in case of Janssen's vaccine) and a booster schedule if she has received three doses (or two doses in case of Janssen's vaccine).

## Analysis of the risk associated with vaccination for the occurrence of the neonatal events of interest through an exploration of case–controls nested in a cohort

A matched case–control approach nested in a cohort will be carried out for each of the events of interest.

### Cohort definition

► Pregnant people (preterm to late term) and with mother's age data.
► Live births with delivery date available based on the best available obstetric estimator.
► Perinatal medical record with 'COVID-19 vaccine module' implemented.

### Case definition (events of interest)

1. Premature. Preterm delivery: live birth at less than 37 weeks gestational age according to the best available obstetric estimator.
2. Small for gestational age: live birth classified as small based on country standards.
3. Low birth weight: born alive weighing less than 2500 g.

### Definition of controls

Two groups of three (uneventful) controls will be randomly selected for each case. The first group of controls will be randomly selected at the same delivery date of each case matched by delivery hospital and mother's age (±3 years). The second group of controls will be births randomly selected from the estimated conception time of the cases based on the estimated delivery date (EDD). EDD will be estimated according to the best available estimator considering the following characteristics: (1) early ultrasound and (2) clinical (LMP). If EDD is not available, gestational age considered during newborn examination should be use. Second group of controls will

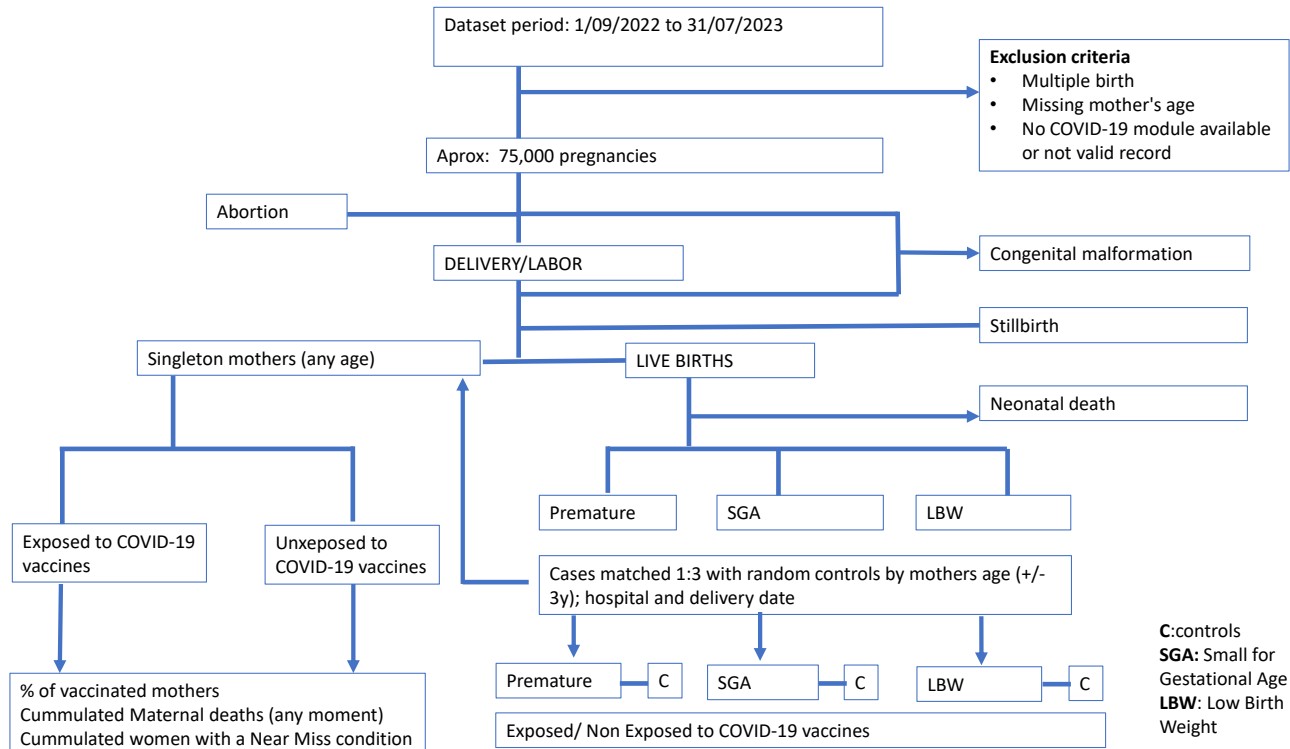

**Figure 1** Overview of the study procedure for data collection and outcomes of interest.

be also matched by delivery hospital and mother's age (±3 years).

### Definition of exposure

Vaccines considered will be those mentioned in the Exposure of interest (immunisation with COVID-19 vaccines) section.

Exposure shall be classified according to the following exposure windows:

1. 'Late immunisation': pregnant people immunised on the index date or up to one trimester before the end of pregnancy (third trimester).
2. 'Second trimester immunisation': pregnant people immunised after completing the first trimester until the second trimester before the event.
3. 'First trimester immunisation': pregnant people immunised during the first trimester.
4. 'Past immunisation': immunised prior to pregnancy
5. 'Immunisation at any time': immunisation at any of the above times.

Moreover, gestational age (weeks) will be considered if available (online supplemental material 2)

Figure 1 summarises the overall conceptual framework and procedures of this study.

### Expected sample size and rationale for its determination
### General considerations

In estimating the sample size, the frequencies of events observed in a previous analysis in the SIP-PLUS database were considered to estimate the number of potential events (online supplemental material 3). Given the lack of precise data on vaccination coverage in hospitals in the CLAP network, literature values from studies published in other countries (prevalences ranging from 20% to 50%) were used as a reference.[27 28] The design of the contrast tests as bilateral (or two tailed) is because the hypothesis contemplates the scenario of a possible protection against adverse events associated with vaccines and also the opposite scenario, with an increased risk. The correlation coefficient (r) between (matched) cases and controls is not known and cannot be estimated from previous studies. Hence, a conservative value of 'r=0.2' has been assumed for the calculation of the sample size, following the recommendation of some authors for these cases.[39] In this part, we will consider the analysis of events for which, a priori, there would be sufficient statistical power, anticipating that the measures of association (OR) may not be to large based on existing published studies.

### Main analysis of vaccination (any vaccine) with respect to 'non-use'

For the matched case–control study (matched with three controls) with a power of 80%, alpha error=0.05 (bilateral), a case–control correlation coefficient of 0.2 and a vaccine exposure of 20% would require at least 1009 cases (3027 controls) to detect an increased probability of vaccine-associated event risk of 30% (OR=1.3) and at least 650 cases (1950 controls) to detect 30% protection (OR=0.7) (figures 2 and 3).

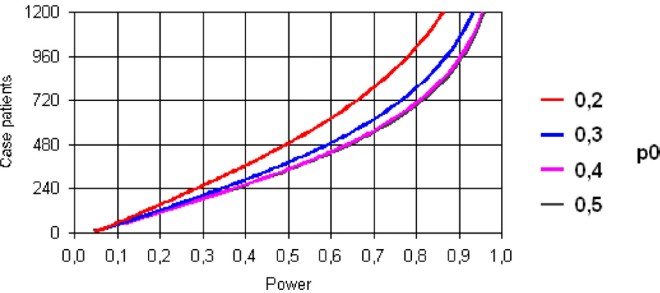

**Figure 2** Power of the analysis as a function of the number of cases for different probabilities of exposure to vaccines from 20% to 50% for the detection of an OR=1.3 (increased risk).

### Estimation of the expected number of cases in the database

Based on previous estimates in the SIP-PLUS database and the annual number of deliveries in each hospital of the network, enough cases could be expected annually: 5731 cases of small for gestational age, 10 932 preterm and 5995 low birthweight babies.

### Data analysis

Retrospective description (cumulative incidence) of the following maternal and neonatal events in the pregnant cohort: preterm, small for gestational age, low birth weight, stillbirth, neonatal death, congenital malformations, MNM and maternal death

The cumulative incidence will be estimated together with the 95% CI using the exact method. When calculating the cumulative incidence, the number of events of interest will be used in the numerator and different denominators according to the type of event.

1. 'Live births' denominator: small for gestational age, premature, low birth weight, stillbirth, MNM and maternal death.
2. 'Total pregnancies' denominator: stillbirth, congenital malformations.

### Description of the proportion of pregnant people who have received a dose of COVID-19 vaccine (total and by type of vaccine) and complete vaccination schedule

The proportion of vaccinated pregnant people out of the total number of pregnant people in the database will be estimated. The proportions (prevalences) of immunisation will be estimated by type of vaccine, vaccine platform (ARNm, Adenoviral vector based), schedule (complete/

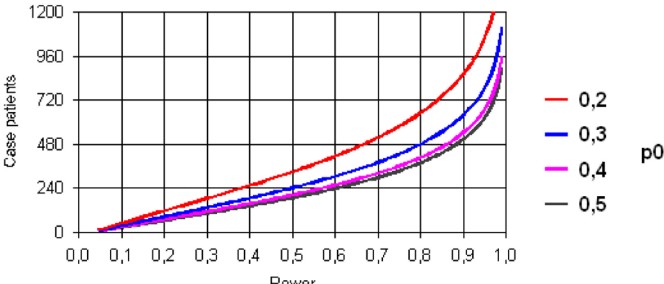

**Figure 3** Power as a function of the number of cases for different probabilities of exposure to vaccines from 20% to 50% for the detection of an OR=0.7 (protection).

incomplete) and time of administration (prepregnancy, first trimester, second trimester, third trimester). The total prevalence of maternal immunisation (combination of all vaccines administered at any time) will also be estimated. The 95% CI of the proportion will be calculated using the exact method.

### Nested case–control analysis

Logistic regression models will be built for each event of interest, considering COVID-19 vaccine exposure as the main independent variable adjusting for possible confounders (covariates) and estimating the OR together with its 95% CIs as a measure of association (OR). The use of a 'standard' (non-conditional conditional) logistic regression analysis will consider adjustment for potential confounders and in addition to adjustment variables, but in addition matched variables will be included in the adjusted models, as recommended for the minimisation of bias in the analysis of matched case–control studies.[40] Models will include: (1) unadjusted models with only the matching variables (mother's age, date of delivery or conception time, and hospital) and (2) a fully adjusted models that also includes all possible confounders described above. The case–control analyses will follow two approaches depending on the control group: (1) approach A will consider the randomly selected sample of controls from deliveries matched on the same birth date, maternal age and attended in the same hospital and (2) approach B will consider the controls randomly selected based on the best estimate of the conception time (LMP or early ultrasound and the due date), same maternal age and attended in the same hospital (online supplemental material 4). As a rule, missing variable data will be treated as a separate (unknown) category. If necessary, missing data will be handled by multiple imputation methods.

Analyses by individual vaccine type or platform will be explored. Sensitivity analysis will be performed for different exposure time windows considering the different types of exposure considered in the paper, and their classification according to proximity to the event (late vaccination, second trimester, early), completeness of the schedule (complete or incomplete schedule) and country. Sample size estimation were made using PS: Power and Sample Size Calculation (V.3.1.6).[41] Statistical analyses will be performed using Stata V.17 statistical software (Stata).

### Patient and public involvement

This protocol presents a retrospective register-based, non-experimental study and will not have measures of intervention that could burden patients in any way assessable not constituting any risk for patients' participation, and it neither implies the realisation of diagnostic tests or additional therapeutic decision besides of what the professional considers timely necessary as part of their clinical practice. In addition, the research question formulation was based on our own priorities for patient benefit and result interpretation. The nested case–control study

described by the protocol is based on clinical records from a network of hospital sites and thus will not include a recruitment process for patients. There was no patient or public involvement in the design of the current study protocol nor conduct of the study. However, results will be distributed to pregnant people groups, as well as peer-reviewed journals and scientific conferences.

## Strengths and limitations

Risk of indication bias: It is possible that women at higher baseline risk for severe COVID-19 may be more likely to have been vaccinated than those not presenting these factors. Adjusted models will consider inclusion of risk factors (eg, diabetes, hypertension, body mass index $\geq 30 \, \text{kg/m}^2$) as adjustment variables.[4 42]

Availability of controls: The number of controls available for case-matching may be small. In field studies, obtaining controls is very expensive and getting the desired number may turn out unfeasible. However, in database studies, controls are more readily available. Nevertheless, in the exceptional case of lack of sufficient controls. A contingency plan will be carried out considering the following scenarios. Scenario 1: widening the controls' maternal age to ±5 years (keeping the same hospital and delivery date or due date) and scenario 2: delivery date or due date to 1 week (maintaining maternal age ±3 and same hospital). Since we will use two different approaches for control selection, the analysis from both perspectives will allow to test the consistency of the results.

Possible lack of 'non-exposure' to COVID-19 vaccines in pregnant people (case of massive recommendation for maternal immunisation and full coverage). We understand that current recommendations are aimed at promoting maternal immunisation with COVID-19 vaccines, but full coverage is not anticipated at present. However, an interim analysis is planned at the end of the year to assess the prevalence of exposure by that time, and to readjust the initial estimates of sample size and study potential before the start of the study.

Missing data: As in any study with secondary data, there are always limitations due to incomplete data records, inaccuracies (dates), unmeasured confounding factors and missing data, etc. In case of missing data for a variable, these will be treated as a separate category in the analysis, and multiple imputation techniques will be considered whenever necessary.

Moreover, the description of abortions will be considered exploratory/hypothetical since: (1) it is not possible to differentiate whether the abortions recorded are spontaneous or induced and (2) a significant proportion of spontaneous abortions occurs on the first weeks of pregnancy and they may not require hospital care nor attended in maternal hospital settings.

Finally, randomised clinical trials (RCTs) are the gold standard for causal inference. However, comparative observational studies may be appropriate when RCTs are not feasible. With the new recommendations for maternal immunisation issued by the CDC and WHO RCTs comparing COVID-19 vaccines versus placebo would be difficult to carry out since clinical equipoise will not be reached and administering placebo would be considered unethical. In this scenery, retrospective observational studies can provide comparative evidence on COVID-19 vaccine effects versus unexposed pregnant people. Furthermore, information on COVID-19 vaccine safety in pregnant people might mainly be coming, as already mentioned, from high-income countries (HICs). This study has the strength of generating evidence of COVID-19 vaccines which are those currently being used in Latin American and Caribbean, asides from mRNA. It represents an opportunity to produce complementary evidence of maternal, fetal and neonatal adverse events from hospitals in middle-income and low-income settings but also to gather information for vaccines whose safety profile in pregnancy is less known such as vaccines non authorised nor used in HICs.

## ETHICS AND DISSEMINATION

The present proposal will be implemented in compliance with the principles of the Declaration of Helsinki[43] and the international standards for the conduct of epidemiological studies, contained in the International Guidelines for Ethical Review of Epidemiological Studies.[44] The reporting of results will conform to the guidelines for reporting routinely collected and observational pharmacoepidemiology health data (RECORD-PE).[45] The study protocol was reviewed by the Ethical Review Committee on Research of the Pan American Health Organization. Patient informed consent was waived due to the retrospective design of the study and the utilisation of anonymised data (Ref. No: PAHOERC.0546.01). Study results will be disseminated in open access journals.

### Benefit–risk assessment for participants

This retrospective study based on secondary data does not entail any additional risk for the patients because of their participation. Nor does it involve additional diagnostic tests, evaluation or therapeutic decisions beyond those deemed appropriate as part of routine clinical practice. However, the results of the proposal are expected to inform the understanding of the benefit–risk balance of maternal immunisation against COVID-19 in the future.

## Confidential treatment of data

As the study will be carried out on an anonymised database without personal identifiers, an informed consent waiver was granted.

### Author affiliations

[1]Department of Medicine, University of Valencia, INCLIVA Health Research Institute and CIBERSAM, Valencia, Spain
[2]Fundación para la innovación, la formación, la investigación y el desarrollo comunitario (FUNDEC), Santa Cruz de Tenerife, Spain
[3]Unidad de Investigación Clínica y Epidemiológica Montevideo, Montevideo, Uruguay
[4]Deparment of Preventive and Social Medicine, School of Medicine, Montevideo, Uruguay
[5]Innovation, Access to Medicines and Health Technologies (IMT), PAHO, Washington, District of Columbia, USA
[6]Department of Obstetrics and Gynecology, Pereira Rossell Hospital, School of Medicine, Universidad de la Republica Uruguay, Montevideo, Uruguay
[7]PAHO, Washington, District of Columbia, USA
[8]Latin American Center of Perinatology, Women and Reproductive Health (CLAP/WR), PAHO, Montevideo, Uruguay
[9]Centro de Estudios de Estado y Sociedad (CEDES), Buenos Aires, Argentina

**Correction notice** This article has been corrected since it was published. The name of the author Edgard Rojas has been corrected to Robin Rojas-Cortés.

**Contributors** DMS-G, AMR, BDM, CS, JLC, SJS and MColomar wrote the proposal. DP, MChiu, MV-G, RR-C, EA, PD and RGPdL reviewed the proposal and made remarkable contributions. GT, LM and MR designed the clinical records. All the authors reviewed and agree with the protocol.

**Funding** This study is funded by Health Canada: Grant Number: 452029: 'Access to COVID-19 Vaccines and Therapeutics for Populations in Situations of Vulnerability in the Americas and to support maternal immunisation safety related activities'; US Food and Drug Administration grant to support maternal immunisation safety related activities, and by the US government (USG) funds through the American Rescue Plan Act.

**Disclaimer** The author is a staff member of the Pan American Health Organization. The author alone is responsible for the views expressed in this publication and they do not necessarily represent the views, decisions or policies of the Pan American Health Organization.

**Competing interests** The authors declare that they have no known competing financial interests or personal relationships that could have appeared to influence the work reported in this paper.

**Patient and public involvement** Patients and/or the public were not involved in the design, or conduct, or reporting, or dissemination plans of this research.

**Patient consent for publication** Not applicable.

**Provenance and peer review** Not commissioned; externally peer reviewed.

### ORCID iDs

Diego Macías Saint-Gerons http://orcid.org/0000-0002-2572-2160
José Luis Castro http://orcid.org/0000-0002-9179-113X
Mercedes Colomar http://orcid.org/0000-0001-7424-5551
Robin Rojas-Cortés http://orcid.org/0000-0002-4046-9298
Claudio Sosa http://orcid.org/0000-0002-2539-0848
Suzanne Jacob Serruya http://orcid.org/0000-0003-1371-4558
Edgardo Abalos http://orcid.org/0000-0001-6653-429X
Pablo Durán http://orcid.org/0000-0003-1032-487X
Rodolfo Gomez Ponce de León http://orcid.org/0000-0001-9914-7130
Giselle Tomasso http://orcid.org/0000-0002-4424-1581
Bremen De Mucio http://orcid.org/0000-0003-0662-9742

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
