## [Reviewer comments · BMJ Open]

This paper was submitted to a another journal from BMJ but declined for publication following peer review. The authors addressed the reviewers' comments and submitted the revised paper to BMJ Open. The paper was subsequently accepted for publication at BMJ Open.

ARTICLE DETAILS

TITLE (PROVISIONAL)	Description of maternal and neonatal adverse events in pregnant people immunized with COVID-19 vaccines during pregnancy in the CLAP NETWORK of sentinel sites. Nested case-control analysis of the immunization-associated risk: A study protocol
AUTHORS	Macías Saint-Gerons, Diego; Castro, José; Colomar, Mercedes; Rojas, Robin; Sosa, Claudio; Roperó, Alba; Serruya, Suzanne; Pastor, Desiré; Chiu, Monica; Velandia-Gonzalez, Martha; Abalos, Edgardo; Durán, Pablo; Gomez Ponce de León, Rodolfo; Tomasso, Giselle; Mainero, Luis; Rubino, Marcelo; De Mucio, Bremen

VERSION 1 – REVIEW

REVIEWER	Theiler, Regan Mayo Clinic, Obstetrics and Gynecology
REVIEW RETURNED	19-Mar-2023

GENERAL COMMENTS	This work will be important to add to the safety data for multiple types of COVID-19 vaccines in pregnancy. As stated by the authors, most currently published data addresses safety of mRNA vaccines and the Janssen vaccine. 1. Line 28-- word is missing after "will be."2. It would be helpful to the reader to discuss all vaccines (types, names, etc) used in the subject countries during the timeframe of the proposed cohort study. A table would be helpful for this material.3. Under inclusion criteria, "termination of pregnancy" in the US means induced abortion. I suggest using "delivery" instead for universal shared understanding.
--

REVIEWER	Moro, Pedro L. Centers for Disease Control and Prevention
REVIEW RETURNED	20-Mar-2023

GENERAL COMMENTS	This protocol describes a planned study of maternal and neonatal adverse events in women immunized with COVID-19 vaccines during pregnancy using a nested case-control analysis design. The study is a multisite international study in Latin America. The study covers an important area of study in public health and has the potential to provide valuable information on maternal safety of the COVID-19 vaccines. I recommend its publication. I include
--

	some comments for the authors to clarify a few areas that were not clear. Lines 25- 28 there seem to be something missing at the end of this sentence: "Moreover, the risk of prematurity, small for gestational age, and low birth weight associated with exposure to COVID-19 vaccines will be" Read until page 11. Start in General and specific objectives: The objectives include describing the frequency of preterm delivery, small for gestational age, and low birth in a cohort of pregnant women as well as through a case-control design. The authors indicate they plan to use the SIP-database to identify pregnancies. Please provide and explanation of this electronic database and how will pregnancies be identified and the adverse events? Through ICD-10 codes? Or other coding? Does this electronic database collect data from routine healthcare encounters of these women in the participating hospitals? The authors may want to explain Page 12, line 34, for the stated objective Why so few events will be analyzed? However, later on in page 13, line 41 the authors state adverse events of interest. Methods and analysis Is there a group or team that will conducted the analysis? Or will each country conduct analysis of their own data following an established analytic protocol? Ethics and dissemination It is indicated that the ethics committee of PAHO has reviewed the study. However, since this is an international study with sites in different countries, the ethics committees of the participating countries should have also reviewed the study. Can the authors clarify this?
--	---

REVIEWER	DeSilva, Malini B. HealthPartners Institute for Medical Education, Research
REVIEW RETURNED	22-Mar-2023

GENERAL COMMENTS	The authors provide a study protocol related to monitoring the occurrence of neonatal and maternal outcomes following COVID-19 vaccine during pregnancy. The outcomes need further clarity and overall the authors should provide more information about the study procedures. There are a number of issues with the study methods including that the time period included would allow for maternal immunizations as early as late March 2021. It would be helpful to know about COVID-19 vaccine roll-out in the included countries to help inform the expected % of women vaccinated during, before, or after pregnancy. Additionally, the time period includes a time when bivalent boosters were available, yet there is no mention of this in the protocol. There have been many studies published to date regarding COVID-19 vaccination during pregnancy along with infant and maternal outcomes. The authors should revisit this literature and update their introduction regarding what is currently known. Additionally, there are some specifics that would help with clarity.  - All figures need captions. - The flow diagram is confusing and it is not clear how the cohorts for each outcome are identified. - Throughout "women" is used, consider changing to pregnant persons/people or other similar From Introduction, page 6:
--

	 - Lines 31 - 32 - Please specify "coronavirus disease 2019 (COVID-19)" - Lines 33 - 34 - Rather than saying, "the former women than among the latter," consider changing to "pregnant persons compared with non-pregnant persons." - Lines 25 - 29 - "Mortality in pregnant women with COVID-19 could be 70% higher, and the risk of admission to intensive care units (ICU) or the need for mechanical ventilation would be three times higher compared to non-pregnant women." Please specify if this is for pregnant persons unvaccinated against COVID-19 or any pregnant persons - Lines 40-41 - Rather than saying, "would have," if citing a study showing a higher risk of adverse maternal-fetal outcomes, please state that these outcomes are higher - Lines 48 - 54 - Please clarify if infants born to persons with COVID at any time during pregnancy are at higher risk of these complications or if only at time of delivery - Page 7 - Please revise the first full paragraph for clarity. For instance, "It should be noted that a characteristic of the new COVID-19 vaccines is that they were developed from a range of different platforms: viral vector (adenoviral), mRNA platform, inactivated, attenuated viruses, being pregnancy an exclusion criterion in the main pivotal clinical trials undertaken for the authorization of these COVID-19 vaccines." I think what you want to say is that pregnancy was an exclusion criteria for the main clinical trials, but this is not clear. - For inclusion criteria, please include date range - For exclusion criteria please note whether any exclusions based on birth outcome (e.g., stillbirth, SAB, etc.); rather than only citing twin gestations as an exclusion, may want to include "multiples" - For adverse events of interest, please specify for #9 whether this includes both spontaneous abortion and therapeutic abortion categories - For obstetric variables, is there a measure of prenatal care that can be included? - For obstetric variables, are diabetes and hypertension pre-existing or gestational? Please clarify - Section 3.2.1 - it is unclear what "pregnant women at term" means given that pre-term birth is an outcome under study - Section 3.2.3 - definition of controls - matching on delivery date does not allow for controlling by gestational age which is important for many of the outcomes. Given the multiple outcomes, different cohorts may be necessary to create an appropriate control group for a given outcome - Section 3.2.4 - Please include gestational weeks during which immunization may occur for each of the immunization categories - Additional references to include regarding what is already known about safety of COVID-19 vaccines in pregnancy:  1) DeSilva MB, Haapala J, Vazquez-Benitez G, et al. Evaluation of Acute Adverse Events after Covid-19 Vaccination during Pregnancy. N Engl J Med. 2022 Jul 14;387(2):187-189. doi: 10.1056/NEJMc2205276. Epub 2022 Jun 22. PMID: 35731916; PMCID: PMC9258750. 2) Lipkind HS, Vazquez-Benitez G, DeSilva M, Vesco KK, Ackerman-Banks C, Zhu J, Boyce TG, Daley MF, Fuller CC, Getahun D, Irving SA, Jackson LA, Williams JTB, Zerbo O, McNeil
--	--

	MM, Olson CK, Weintraub E, Kharbanda EO. Receipt of COVID-19 Vaccine During Pregnancy and Preterm or Small-for-Gestational-Age at Birth - Eight Integrated Health Care Organizations, United States, December 15, 2020-July 22, 2021. MMWR Morb Mortal Wkly Rep. 2022 Jan 7;71(1):26-30. doi: 10.15585/mmwr.mm7101e1. PMID: 34990445; PMCID: PMC8735559. 3) Ibroci E, Liu X, Lieb W, Jessel R, Gigase FAJ, Chung K, Graziani M, Lieber M, Ohrn S, Lynch J, Castro J, Marshall C, Tubassum R, Mutawakil F, Kaplowitz ET, Ellington S, Molenaar N, Sperling RS, Howell EA, Janevic T, Dolan SM, Stone J, De Witte LD, Bergink V, Rommel AS. Impact of prenatal COVID-19 vaccination on delivery and neonatal outcomes: Results from a New York City cohort. Vaccine. 2023 Jan 16;41(3):649-656. doi: 10.1016/j.vaccine.2022.09.095. Epub 2022 Dec 14. PMID: 36526507; PMCID: PMC9749885.
--	--

VERSION 1 – AUTHOR RESPONSE

RESPONSE TO REVIEWER’S COMMENTS:

Reviewer #1. Dr. Regan Theiler, Mayo Clinic

This work will be important to add to the safety data for multiple types of COVID-19 vaccines in pregnancy. As stated by the authors, most currently published data addresses safety of mRNA vaccines and the Janssen vaccine.

R1#GenComment

Response to Reviewer 1 – General Comment

Thank you very much for your kind and constructive comments, and for your time to review our manuscript.

R1#1: Line 28-- word is missing after "will be."

Response to Reviewer 1 - Comment #1:

Thank you very much to note this typo. The word missing at the end is “estimated”. The end of the sentence has been completed accordingly.

R1#2. It would be helpful to the reader to discuss all vaccines (types, names, etc) used in the subject countries during the timeframe of the proposed cohort study. A table would be helpful for this material.

Response to Reviewer 1 - Comment #2:

Thank you very much for the comment. We acknowledge the suggestion of the reviewer to provide a table with with the supply of products in the country. The vaccines of interest including types, names and manufacturers are detailed in the protocol, section 3.1.5 “Exposure of interest”; We preferred not to provide a “static” table with the vaccines per country since the supply of vaccines in a country can change over time due to donations or as new vaccines are incorporated after authorization. In order to provide the supplementary context to the reader we have added a reference that summarize the vaccines available in the countries during the study period on a continuous basis.

Please find the new reference 31. Panamerican Health Organization. Pharmacovigilance for COVID-19 vaccines. Available from: <https://covid-19pharmacovigilance.paho.org>

R1#3. Under inclusion criteria, "termination of pregnancy" in the US means induced abortion. I suggest using "delivery" instead for universal shared understanding.

Response to Reviewer 1 - Comment #3:

Thank you very much for your comment. Following your suggestion, we have changed "termination of pregnancy" for "delivery".

Reviewer #2. Dr. Pedro L. Moro, Centers for Disease Control and Prevention

This protocol describes a planned study of maternal and neonatal adverse events in women immunized with COVID-19 vaccines during pregnancy using a nested case-control analysis design. The study is a multisite international study in Latin America. The study covers an important area of study in public health and has the potential to provide valuable information on maternal safety of the COVID-19 vaccines. I recommend its publication. I include some comments for the authors to clarify a few areas that were not clear.

R2#GenComment

Response to Reviewer 2 – General Comment

Thank you very much for your kind and constructive comments, and for your time to review our manuscript.

R2#1 Lines 25- 28 there seem to be something missing at the end of this sentence: "Moreover, the risk of prematurity, small for gestational age, and low birth weight associated with exposure to COVID-19 vaccines will be"

Response to Reviewer 2 – Comment #1:

Thank you very much to note this typo. The word missing at the end is "estimated". The end of the sentence has been completed accordingly.

R2#2 Read until page 11. Start in General and specific objectives: The objectives include describing the frequency of preterm delivery, small for gestational age, and low birth in a cohort of pregnant women as well as through a case-control design.

The authors indicate they plan to use the SIP-database to identify pregnancies. Please provide and explanation of this electronic database and how will pregnancies be identified and the adverse events? Through ICD-10 codes? Or other coding? Does this electronic database collect data from routine healthcare encounters of these women in the participating hospitals? The authors may want to explain

Response to Reviewer 2 – Comment #2:

We acknowledge your comment. Yes, the objectives included: 1) descriptive objectives: description of describing the frequency of preterm delivery, small for gestational age, and low birth in the cohort of SIP-PLUS; and 2) analytic objectives through a nested case-control in the cohort. Please refer figure 1 for the complete flow chart of the study.

R2#3 The authors indicate they plan to use the SIP-database to identify pregnancies. Please provide and explanation of this electronic database and how will pregnancies be identified and the adverse events? Through ICD-10 codes? Or other coding? Does this electronic database collect data from routine healthcare encounters of these women in the participating hospitals? The authors may want to explain

Response to Reviewer 2 – Comment #3:

Singleton pregnancies can be identified from all the electronic perinatal clinical recorded in the database. The events of interest for each pregnancy are coded events according to the definitions detailed in the section “3.1.3 Adverse events of interest”; These codes used in the electronic database are based in ICD-9 (The database is currently being updated to ICD-11).

This electronic database collect data from routine encounters of these women in the participating hospital through their perinatal clinical records. In summary, it is a healthcare database of perinatal electronic medical records that centralizes the demographic, diagnostic, obstetric and neonatal clinical data of mothers and their offspring. The explanation of the SIP-database has been updated.

R2#4 Page 12, line 34, for the stated objective Why so few events will be analyzed? However, later on in page 13, line 41 the authors state adverse events of interest.

Response to Reviewer 2 – Comment #4:

Thank you very much for your question. The descriptive objectives will be performed using the totality of the cohort (all the events; figure 1). For the analytic objectives, a minimum number of cases (and controls) will be required. Therefore, the events for the analytic approach in nested-case control were selected for its anticipated frequency to make possible that in the study period we have the necessary number of events/cases to carry out the study with sufficient statistical power.

R2#5 Methods and analysis

Is there a group or team that will conduct the analysis? Or will each country conduct analysis of their own data be following an established analytic protocol?

Response to Reviewer 2 – Comment #5:

Thank you. Regarding your question, the statistical analysis will be performed centrally by the team that maintains the database in collaboration with epidemiologists from PAHO Regional Headquarters. This point was clarified in the 5.3 Nested case-control analysis section.

R2#6 Ethics and dissemination

It is indicated that the ethics committee of PAHO has reviewed the study. However, since this is an international study with sites in different countries, the ethics committees of the participating countries should have also reviewed the study. Can the authors clarify this?

Response to Reviewer 2 – Comment #6:

Thank you for your question. This is an observational (non-interventional) retrospective study including anonymized data. The study does not consider additional diagnosis, tests or biological samples. For the above-mentioned reasons, a favorable opinion of a single Ethics Committee along with the approval of the hospital’s director, or the “Teaching and research committee” or “Ethics committee” when available at the hospitals, were considered.

It is important to note that a possible redundant ethical evaluation of the same protocol by several committees must be disassociated from the necessary agreement to carry out the study with each of the centers.

Reviewer: 3. Dr. Malini B. DeSilva, HealthPartners Institute for Medical Education

The authors provide a study protocol related to monitoring the occurrence of neonatal and maternal outcomes following COVID-19 vaccine during pregnancy. The outcomes need further clarity and overall, the authors should provide more information about the study procedures. There are a number

of issues with the study methods including that the time period included would allow for maternal immunizations as early as late March 2021.

R3#1

Response to Reviewer 3 – Comment #1:

Thank you very much for your kind and constructive comments, and for your time to review our manuscript.

Yes, this is an observational study reflecting the current clinical practice. Therefore, the past period were the first COVID-19 vaccines were authorized is part of the study period. According to the summary of product characteristics of the COVID-19 vaccines there were no data of use during pregnancy but maternal immunization was not a contraindicated. Therefore, in the absence of data on use in the pregnancy, its administration to pregnant women was left to the discretion of the doctor after consensus with the patient, assessing the possible risks and expected benefits on a case-by-case basis. There may have been progress from occasional in the earliest times to a major use recommendation incorporated into the guidelines.

R3#2 It would be helpful to know about COVID-19 vaccine roll-out in the included countries to help inform the expected % of women vaccinated during, before, or after pregnancy. Additionally, the time period includes a time when bivalent boosters were available, yet there is no mention of this in the protocol. There have been many studies published to date regarding COVID-19 vaccination during pregnancy along with infant and maternal outcomes. The authors should revisit this literature and update their introduction regarding what is currently known.

Response to Reviewer 3 - Comment #2:

Thank your comment. Ideally, knowing in advance the detail on vaccination coverage data by country or even by population attended by the hospital would have been desirable. But unfortunately, as happens often in many studies, this information was not available neither from the literature nor from the national authorities. Data from Latin America and the Caribbean on pregnancy exposure use was lacking. Thus, the estimation of the sample size took into account the number of expected events (which can be inferred) and for the vaccination coverage, the most conservative values (low vaccination rate) were taken based on data published in the international literature at the moment the protocol was submitted.

Since the assumptions for the sample size were most conservative and that it is reasonable to think that the exposure is now greater than anticipated, the statistical power of the study will not be penalized but on the contrary, it could be higher than expected.

We acknowledge that evidence on the topic is accumulating rapidly (originated mainly from High Income Countries studies), thus we have updated some of the references

R3#3 Additionally, there are some specifics that would help with clarity.

- All figures need captions.

Response to Reviewer 3 - Comment #3:

Thank you very much for your suggestion. Captions have been included in all figures during submission process into the journal's web page.

R3#4 The flow diagram is confusing and it is not clear how the cohorts for each outcome are identified.

Response to Reviewer 1 - Comment #4:

Thank you very much for your comment. However, we consider that the Figure 1 summarize appropriately the whole study procedures and the included populations.

R3#5 Throughout "women" is used, consider changing to pregnant persons/people or other similar
Response to Reviewer 3 - Comment #5:

We greatly appreciate your observation. We fully agree with that. As suggested, we have changed to person or people and modified the text and the main title accordingly.

*"Pregnant people" was used preferably (<https://www.cdc.gov/coronavirus/2019-ncov/need-extra-precautions/pregnant-people.html>)

R3#6 From Introduction, page 6:

- Lines 31 - 32 - Please specify "coronavirus disease 2019 (COVID-19)"

Response to Reviewer 3 - Comment #6:

Thank you. Following your advice "coronavirus disease 2019 (COVID-19)" is now specified.

R3#7 Lines 33 - 34 - Rather than saying, "the former women than among the latter," consider changing to "pregnant persons compared with non-pregnant persons."

Response to Reviewer 3 - Comment #7:

Thank you. The sentence has been reworded according to your suggestion.

R3#8 - Lines 25 - 29 - "Mortality in pregnant women with COVID-19 could be 70% higher, and the risk of admission to intensive care units (ICU) or the need for mechanical ventilation would be three times higher compared to non-pregnant women." Please specify if this is for pregnant persons unvaccinated against COVID-19 or any pregnant persons

Response to Reviewer 3 - Comment #8:

Thank you. We refer here to non-pregnant unvaccinated people of similar age. We have included this clarification accordingly.

R3#9 - Lines 40-41 - Rather than saying, "would have," if citing a study showing a higher risk of adverse maternal-fetal outcomes, please state that these outcomes are higher

Response to Reviewer 3 - Comment #9:

Thank you. The sentence has been reworded according to your suggestion

R3#10 - Lines 48 - 54 - Please clarify if infants born to persons with COVID at any time during pregnancy are at higher risk of these complications or if only at time of delivery

Response to Reviewer 3 - Comment #10:

Thank you. In order to clarify this aspect, the evidence cited here is driven from studies including persons with COVID diagnosed at different moments during pregnancy. Please find the cited references for further details.

R3#11 - Page 7 - Please revise the first full paragraph for clarity. For instance, "It should be noted that a characteristic of the new COVID-19 vaccines is that they were developed from a range of different platforms: viral vector (adenoviral), mRNA platform, inactivated, attenuated viruses, being pregnancy an exclusion criterion in the main pivotal clinical trials undertaken for the authorization of these COVID-19 vaccines." I think what you want to say is that pregnancy was an exclusion criterion for the main clinical trials, but this is not clear.

Response to Reviewer 3 - Comment #11:

Thank you for the comment. In order to clarify this, it might be thought that the more unknown/novel vaccine platforms could pose a higher risk or uncertainty and hence this a clear justification for the exclusion of pregnant people out of their clinical studies, on the other hand other vaccines were developed from similar platforms than the vaccines currently recommended for administration during pregnancy. We think that the paragraph presents well the idea of: 1) a diversity of vaccines developed from different platforms and 2) a common link for all: the absence of pregnant women in their pivotal trials (exclusion criteria)

R3#12. - For inclusion criteria, please include date range

Response to Reviewer 3 - Comment #12:

Thank you for your comment. In order to represent the whole spectrum of pregnant people ages attended in "real practice" we preferred not to state one "a priori" date range. However, information of the recruitment period was included.

R3#13 - For adverse events of interest, please specify for #9 whether this includes both spontaneous abortion and therapeutic abortion categories

Response to Reviewer 3 - Comment #13:

Thank you for your comment. As mentioned in limitations abortion is considered an exploratory event since spontaneous abortion and therapeutic abortion categories cannot be separated. This is due to local ethical/confidentiality policies in the data recording of this particular event. Following your advice this has been also specified in the list of events of interest.

R3#14 - For obstetric variables, is there a measure of prenatal care that can be included?

Response to Reviewer 3 - Comment #14:

Thank you for your comment. Unfortunately, a measure of "prenatal care" is not available from the electronic records.

R3#15 - For obstetric variables, are diabetes and hypertension pre-existing or gestational?

Response to Reviewer 3 - Comment #15:

Thank you for your comment. Both, for further clarity we have specified this aspect in the protocol. Please find "Obstetric variables and risk factors (definitions)"

R3#16 - Section 3.2.1 - it is unclear what "pregnant women at term" means given that pre-term birth is an outcome under study

Response to Reviewer 3 - Comment #16:

Thanks a lot for the comment. In section 3.2,1 we refer to the cohort for the nested case-control approach including pregnant women from pre-term to early term, full-term, late term and post-term. We have specified this aspect on Section 3.2.1 accordingly

R3#17 - Section 3.2.3 - definition of controls - matching on delivery date does not allow for controlling by gestational age which is important for many of the outcomes. Given the multiple outcomes, different cohorts may be necessary to create an appropriate control group for a given outcome
Response to Reviewer 3 - Comment #17:

Thank you for your comment. Not all the potential confounders should be matching variables. Most of the potential confounders will be adjusted in the analysis. Therefore, gestational age, along with other potential confounders for many of the outcomes (i.e.: diabetes), will be considered as adjusting variables in the logistic regression models to allow an appropriate comparison.

R3#18 - Section 3.2.4 - Please include gestational weeks during which immunization may occur for each of the immunization categories
Response to Reviewer 3 - Comment #18:

Thank you for your comment. The main criteria for exposure classification as in many previous studies will be trimester of COVID-19 vaccination (<https://pubmed.ncbi.nlm.nih.gov/34990445/>). Following your advice gestational weeks at immunization will be considered.

R3#19 - Additional references to include regarding what is already known about safety of COVID-19 vaccines in pregnancy:

- 1) DeSilva MB, Haapala J, Vazquez-Benitez G, et al. Evaluation of Acute Adverse Events after Covid-19 Vaccination during Pregnancy. *N Engl J Med.* 2022 Jul 14;387(2):187-189. doi: 10.1056/NEJMc2205276. Epub 2022 Jun 22. PMID: 35731916; PMCID: PMC9258750.
- 2) Lipkind HS, Vazquez-Benitez G, DeSilva M, Vesco KK, Ackerman-Banks C, Zhu J, Boyce TG, Daley MF, Fuller CC, Getahun D, Irving SA, Jackson LA, Williams JTB, Zerbo O, McNeil MM, Olson CK, Weintraub E, Kharbanda EO. Receipt of COVID-19 Vaccine During Pregnancy and Preterm or Small-for-Gestational-Age at Birth - Eight Integrated Health Care Organizations, United States, December 15, 2020-July 22, 2021. *MMWR Morb Mortal Wkly Rep.* 2022 Jan 7;71(1):26-30. doi: 10.15585/mmwr.mm7101e1. PMID: 34990445; PMCID: PMC8735559.
- 3) Ibroci E, Liu X, Lieb W, Jessel R, Gigase FAJ, Chung K, Graziani M, Lieber M, Ohrn S, Lynch J, Castro J, Marshall C, Tubassum R, Mutawakil F, Kaplowitz ET, Ellington S, Molenaar N, Sperling RS, Howell EA, Janevic T, Dolan SM, Stone J, De Witte LD, Bergink V, Rommel AS. Impact of prenatal COVID-19 vaccination on delivery and neonatal outcomes: Results from a New York City cohort. *Vaccine.* 2023 Jan 16;41(3):649-656. doi: 10.1016/j.vaccine.2022.09.095. Epub 2022 Dec 14. PMID: 36526507; PMCID: PMC9749885

Response to Reviewer 3 - Comment #19:

Many thanks for the additional references. The provided references have been included and are now cited in the protocol.

VERSION 2 – REVIEW

REVIEWER	DeSilva, Malini B. HealthPartners Institute for Medical Education, Research
REVIEW RETURNED	10-Aug-2023
GENERAL COMMENTS	The authors plan to use a database to retrospectively describe COVID-19 vaccine utilization by pregnant people in hospitals and evaluate association with specific selected maternal and neonatal

	events. The manuscript describes the plans for a nested case-control analysis. The study has the potential to expand what is currently known about the safety of COVID-19 vaccines administered during pregnancy to include different types of COVID-19 vaccines aside from mRNA only. It should be made more clear that this is one of the strengths. The major issue that remains is a methodologic concern. The decision to match on date of delivery is inappropriate for some outcomes such as preterm birth and SGA, these should be matched based on LMP or EDD instead to account for gestational age at outcome. Selecting based on delivery date biases the sample to select more preterm births at the end of the study period and longer gestations at the beginning of the study period. It may be beneficial for the authors to read the following AJE article on reducing bias in observational studies of vaccination in pregnancy: https://academic.oup.com/aje/article/184/3/176/2576150
--	---

VERSION 2 – AUTHOR RESPONSE

Reviewer: 3

Dr. Malini B. DeSilva, HealthPartners Institute for Medical Education

Comments to the Author:

RESPONSE TO REVIEWER’S COMMENTS:

Reviewer: 3. Dr. Malini B. DeSilva, HealthPartners Institute for Medical Education

R3#1

The authors plan to use a database to retrospectively describe COVID-19 vaccine utilization by pregnant people in hospitals and evaluate association with specific selected maternal and neonatal events. The manuscript describes the plans for a nested case-control analysis. The study has the potential to expand what is currently known about the safety of COVID-19 vaccines administered during pregnancy to include different types of COVID-19 vaccines aside from mRNA only.

Response to Reviewer 3 – Comment #1:

Thank you very much. We agree with the possibility that the study could bring information from the whole spectrum of vaccines that are being used in Latin America and the Caribbean and not mRNA vaccines exclusively.

Lines have been included in the section “Strengths and Limitations”

This study has the strength of generating evidence of COVID 19 vaccines which are those currently being used in LAC, besides from mRNA. It represents an opportunity to produce complementary evidence of maternal, foetal and neonatal adverse events from hospitals in middle- and low-income settings but also to gather information for vaccines whose safety profile in pregnancy is less known such as vaccines non authorized nor utilized in HICs

R3#2

The decision to match on date of delivery is inappropriate for some outcomes such as preterm birth and SGA, these should be matched based on LMP or EDD instead to account for gestational age at outcome. Selecting based on delivery date biases the sample to select more preterm births at the end of the study period and longer gestations at the beginning of the study period. It may be beneficial for the authors to read the following AJE article on reducing bias in observational studies of vaccination in pregnancy:

<https://academic.oup.com/aje/article/184/3/176/2576150>

Response to Reviewer 3 – Comment #2:

Thank you very much for your comment and for the reference. Thank you also for the opportunity to expand our explanation of the study methodology.

We have read with great interest the reference provided by the reviewer (<https://academic.oup.com/aje/article/184/3/176/2576150>). In their article, Vazquez-Benitez et al., analyze the risk of certain neonatal adverse events associated with maternal immunization (vs non-immunization) identifying some potential biases in comparative cohort studies but no considerations are available regarding matching appropriateness in nested case-control designs, which is the design described in our protocol.

However, based on your suggestion we are including in the study a second control group matched with each case by conception time (based on LMP or EDD). We acknowledge the fact that it might be possible that different environmental factors could vary especially during a pandemic-, and affect the periconceptional moment and the course of pregnancy.-. In this case, not only the virus's transmissibility but also health control measures, the general population's behavior, and even vaccination guidelines, might be important factors affecting differently those periods. Therefore, we'll be considering an additional approach for the selection of the control belonging to the same source population (at conception) of the cases. If this scenario were true, the selection of a control based on the best available estimate (early ultrasound or LMP), would define a better reference group than the selection of a control based on the birth's date of delivery. However, the precise timing of conception could be challenging to estimate.

(<https://pubmed.ncbi.nlm.nih.gov/12762085/>) might explain why the birth date is sometimes preferred for matching purposes in previous analyses in the literature using matched case-control designs (please find below a rapid systematic literature review that we have performed on the topic). We anticipate that these two scenarios of analysis might not yield different results, but we consider that the analysis from both perspectives will be of interest and will allow a better analysis of the consistency of the results and enrich the discussion.

A figure illustrating the strategy for the selection of the controls in the case-control approaches has been added (Supplemental material IV)

Accordingly, we have modified the manuscript as follows:

1) A paragraph has been added in the section “3.2.3 Definition of controls”

“Two groups of three (uneventful) controls will be randomly selected for each case. The first group of controls (A) will be randomly selected at the same delivery date of each case matched by delivery hospital and mother's age (+/- 3 years). The second group of controls (B) will be births randomly selected from the estimated conception time of the cases based on the estimated delivery date (EDD). EDD will be estimated according to the best available estimator considering the following characteristics: 1) Early ultrasound; 2) Clinical (last menstrual period –LMP-). If EDD is not available, gestational age considered during newborn examination should be considered. Controls will be also matched by delivery hospital and mother's age (+/- 3 years)”

2) Paragraph has been added in the section “5.3 Nested case-control analysis”

“The case control analyses will follow two approaches depending on the control group: Approach A will consider the randomly selected sample of controls from deliveries matched on the same birth date, maternal age and attended in the same hospital and 2) Approach B will consider the controls randomly selected based on the best estimate of the conception time (LMP or early ultrasound and the due date), same maternal age and attended in the same hospital”

3) Lines have been included in the section “Strengths and Limitations”

“Availability of controls. The number of controls available for case-matching may be small. In field studies, obtaining controls is very expensive, and getting the desired number may not be feasible. However, in database studies controls are more readily available. Nevertheless, in the exceptional case of a lack of sufficient controls, a contingency plan will be carried out considering the following

scenarios. Scenario 1: widening the controls' maternal age to +/- 5 years (keeping the same hospital and delivery date); scenario 2: delivery date to one week (maintaining maternal age +/- 3 and the same hospital). Since we will use two different approaches for control selection, the analysis from both perspectives will allow us to test the consistency of the results"

A table including a Rapid review of case-control studies in the literature evaluating the risk of SGA, LWB or PTB is included in the attached file.

However, in the case of Small for Gestational Age babies, we do not think that baseline differences mentioned in the article can be a source of bias to be considered in this observational study design. In this matched nested case-control approach, the potential of confounding because of an imbalance of baseline risk factors is addressed through two strategies: 1) by design (matching variables) and 2) by analysis, adjusting variables included in the multivariate statistical models.

Potential confounders identified by Vazquez-Benitez are also addressed in this protocol such as sociodemographic variables including race/ethnicity, maternal age at date of delivery, in addition to other factors such as marital status and educational level (none, elementary, high school, college, university) that we also consider important. Similar to the study by Vazquez-Benitez et al., medical conditions such as diabetes or hypertension will be also considered in the protocol. Additionally, other obstetric potentially important variables examined by other studies and not by Vazquez-Benitez such as parity will be considered in the protocol as well, (Please find section 3.1.4 covariates).

VERSION 3 – REVIEW

REVIEWER	DeSilva, Malini B. HealthPartners Institute for Medical Education, Research
REVIEW RETURNED	31-Oct-2023
GENERAL COMMENTS	Thank you for taking the time to create a new control group and think through the pros and cons of each control group. I think the comparison of results based on control group may be able to be a separate methodology manuscript for your group to consider writing.